# First report of MDR virulent *Pseudomonas aeruginosa* in apparently healthy Japanese quail (*Coturnix japonica*) in Bangladesh

**Alamgir Hasan**[1☉]**, Md. Tanjir Ahmmed**[1☉]**, Bushra Benta Rahman Prapti**[1]**, Aminur Rahman**[1]**, Tasnim Islam**[1]**, Chandra Shaker Chouhan**[2]**, A. K. M. Anisur Rahman**[2]**, Mahbubul Pratik Siddique**[1] *

**1** Department of Microbiology and Hygiene, Mymensingh, Bangladesh, **2** Department of Medicine, Bangladesh Agricultural University, Mymensingh, Bangladesh

☉ These authors contributed equally to this work.
* mpsiddique@bau.edu.bd

## Abstract

*Pseudomonas aeruginosa (P. aeruginosa)* is a major pathogen associated conditions like septicaemia, respiratory disorders, and diarrhoea in poultry, particularly in Japanese quail (*Coturnix japonica*). The infection causes huge economical losses due to its high transmissibility, mortality and zoonotic potential. This study aimed to isolate, identify, detect virulence genes, and profile multidrug resistance (MDR) of *P. aeruginosa* from Japanese quail. Oral and rectal swabs were collected from 110 apparently healthy quail birds across various districts in Bangladesh. Bacterial isolation and identification were performed using cultural, morphological, biochemical, and polymerase chain reaction (PCR) methods. Antibiotic susceptibility was assessed using the disc diffusion method, and virulence genes were detected through PCR. Multivariable logistic regression was used to identify risk factors for *P. aeruginosa* infection. Both conventional and PCR methods revealed that 25 (22.73%) of the quail birds were positive for *P. aeruginosa*. The results showed that quail birds in Narsingdi were five times more likely to harbor *Pseudomonas* species (OR: 5.02; 95% CI: 1.34–18.84) compared to those in Mymensingh Sadar. Additionally, quail birds younger than eight weeks had nearly six times higher odds (OR: 5.93; 95% CI: 1.96–17.91) of carrying *Pseudomonas* compared to older birds. Female quail birds had almost four times higher odds (OR: 3.77; 95% CI: 1.30–10.93) of harboring *Pseudomonas* species than males. All 25 *P. aeruginosa* isolates exhibited multi drug-resistance (MDR) patterns. Virulence gene analysis revealed the consistent presence of *exo*A and *rhl*R in all isolates, while *exo*S, *exo*Y, *rhl*I, and *rhl*AB showed variable distribution. The high prevalence of MDR and virulent *P. aeruginosa* in apparently healthy quail birds particularly in Mymensingh and Dhaka divisions, highlights the urgent need for a comprehensive 'One Health' approach to proactively address and mitigate the potential risk this organism poses to both quail and human populations.

**Data Availability Statement:** The data generated during this study is included in the paper and its Supporting Information files.

**Funding:** The author(s) received no specific funding for this work.

**Competing interests:** CONFLICT OF INTEREST 19th September 2024 Title: First Time Detail Virulence and antibiogram profiling of Pseudomonas aeruginosa isolated from apparently healthy Japanese quail (Coturnix japonica) in Bangladesh DECLARATION AND STATEMENT We know of no conflicts of interest associated with the publication. We declare that this manuscript is original and has not been submitted to anywhere else for publication. The manuscript nicely falls under the scope of your journal. Sincerely yours, Mahbubul Pratik Siddique, PhD Corresponding author and Professor Department of Microbiology and Hygiene Bangladesh Agricultural University Mymensingh-2202 Cell Phone: +880-1751-945123 Email: mpsiddique@bau.edu.bd https://www.researchgate.net/profile/Mahbubul-Siddique https://scholar.google.com/citations?user=YmxVEzEAAAAJ&hl=en

## Introduction

Since early 1980s, poultry farming has been emerged as an industry in Bangladesh, and the majority of these poultry are indigenous chickens and ducks, which showed no satisfactory performance [1]. Apart from native poultry, people living in rural and semi-urban areas want a particular kind of bird that can be raised with minimal effort and yield greater financial returns in a short amount of time [2]. The Japanese quail (*Coturnix japonica*) has various characteristics for profitable farming, such as, higher growth rate, earlier sexual maturity, higher rate of egg production, close succession (3–4 generations per year) and less hatching period [3]. Moreover, due to small size and stout body configuration, it attributes to the less feed cost, less space requirements and most importantly, less vulnerability to common infectious diseases [3]. High potential exists for the development of Japanese quail eggs as a less expensive source of protein, particularly in developing nations [4].

The sustainability of quail farming, like that of other poultry, relies on effective management, breeding, feeding, and the implementation of disease prevention, control, and treatment measures. Japanese quail are susceptible to a variety of bacterial infections [5]. Colibacillosis, colisepticemia, fowl cholera, fowl typhoid, and salmonellosis are bacterial illnesses that have been observed in Japanese quail [6]. However, *Pseudomonas* infection in Japanese quail has been looked for unknown reason [7]. Martinez-Laorden et al. (2023) reported *Pseudomonas* spp. as the most prevalent bacterium in Japanese quail [8]. Again, *Pseudomonas aeruginosa (P. aeruginosa)* is the most prevalent species of the genus *Pseudomonas* that impact quails, leading to significant financial losses, elevated morbidity, and elevated mortality rates [9,10].

The bacterium *P. aeruginosa*, a Gram-negative, motile, opportunistic bacterium, is prevalent in the hospital and other environments and it is a normal component of the flora of both immune compromised hosts and healthy people [11]. It causes nosocomial and ventilator-associated infections with a high mortality [12]. *P. aeruginosa* possesses various virulence factors like Pili, exoenzyme S (encoding gene *rhl*I), LasB elastase, exotoxin A (encoding gene e*xo*A), and Phospholipase C, Rhamnolipids (encoding genes *rhl*R, *rhl*AB*)*, whereas the initiation of chronic infections involves alginate and siderophores [13]. Among the five protein secretion systems (type I, II, III (ExoS, ExoY), V, and the latest type VI), *P. aeruginosa* frequently employs these systems to release proteinaceous virulence factor [14,15]. Notably, the type III secretion system enables the direct injection of exotoxins into host cells upon contact, subverting host cell defences and signalling pathways [16]. Bifunctional proteins ExoT and ExoS, possessing ADP-ribosyl transferase and GTPase activating domains, impact the assembly of the actin cytoskeleton in response to extracellular signals by acting on Ras-related GTP-binding proteins like Rho, Rac, and Cdc42 [17]. Additionally, Exotoxin A inhibits protein synthesis by ADP-ribosylating eukaryotic elongation factor 2 [18]. ExoU exhibits phospholipase activity, rupturing eukaryotic membranes upon entering the cytoplasm, while ExoY possesses adenylate cyclase activity [19]. Moreover, *P. aeruginosa* utilizes quorum sensing (QS), a cell-density-dependent mechanism, to regulate the production of various virulence factors [20]. The mechanisms of disease occurrence also involve the bacterium's ability to form biofilms, which contribute to the persistence of *P. aeruginosa* in various environments, making eradication challenging [21,22]. Infections caused by *P. aeruginosa* include pneumonia, urinary tract, wound and soft tissue infections.

Multidrug resistance (MDR) has been increasing globally, posing a significant public health threat due to irrational use of antibiotics, growth promoter and hormones in the livestock sector, particularly in poultry [23,24]. Often, farmers sell birds without considering the recommended withdrawal periods of these drugs, resulting in drug residues remaining in the food chain from animal products [25–27]. This can ultimately lead to the development of MDR in

humans through horizontal gene transfer, selective pressure, gene transfer mechanisms, and ecosystem interactions [25,28]. Recent investigations have highlighted the emergence of multi-drug-resistant bacterial pathogens from diverse sources, emphasizing the necessity for judicious antibiotic use. Furthermore, routine antimicrobial susceptibility testing is crucial for identifying appropriate antibiotics and screening for the emergence MDR strains [21,29–33], as the preventive measures for infectious diseases are largely lacking, and the predominant approach to control such diseases relies heavily on antimicrobial therapy [34]. However, it is crucial to recognize that AMR poses a significant public health emergency in both human and veterinary medicine [35]. The widespread overuse of antibiotics, used for growth-promotion, metaphylactic, and prophylactic purposes in both human medicine and animal husbandry, has led to the emergence and spread of antibiotic-resistant bacteria and resistance genes. This exacerbates the risks to public health and the environment [21,36]. The impact of antimicrobial resistance includes treatment failures, resulting in economic loses, and the potential to serve as a reservoir of resistant bacteria. Therefore, the selection of appropriate antimicrobial agents should be a top priority, emphasizing the need for judicious and responsible use to mitigate the adverse consequences associated with antimicrobial resistance. Identifying disease-causing agents in commercial quail farms is crucial [37] for developing sustainable farming practices [5]. While conventional methods (cultural, morphological and biochemical) are important for bacterial detection, confirmatory diagnosis necessitates the use of polymerase chain reaction (PCR) and virulence gene detection. Existing reports on quail diseases in Bangladesh primarily focus on clinical signs and histopathology [5], and antimicrobial resistance [18], with a notable gap in molecular-based studies for agent and virulence gene detection.

Therefore, this research aims to isolate and identify *P. aeruginosa* from Japanese quails, identify risk factors, detect virulence genes and antibiotic resistance in the bacterial isolates.

## Materials and methods

### Ethical statement

All the animal experimental procedures were approved by the Ethical Approval Committee, Bangladesh Agricultural University, Mymensingh-2202. All the animals were handled during sample collection in compliance with the local animal welfare regulations and maintained according to standard protocols. The approval number is "AWEEC/BAU/2023(73), Date: 24.12.23.

### Sample collection

Samples were collected from apparently healthy live birds in two commercially recognized poultry divisions belonging to Dhaka (Narsingdi and Gazipur districts) and Mymensingh (Mymensingh and Jamalpur districts). A total of 220 samples (110 orals and 110 cloacal swabs) were collected in Luria Bertani (LB) broth from 110 apparently healthy quails from different quail shops (quail selling points) situated in Mymensingh Sadar [n = 60 (30 male + 30 female)], Jamalpur [n = 20 (10 male + 10 female)], Narsingdi [n = 20 (10 male + 10 female)], and Gazipur [n = 10 (5 male +5 female)].

### Bacterial strains, media and growth

The primary enrichment was made in Luria-Bertani (LB) broth (Hi-Media, India), Nutrient broth (Hi-Media, India), and Brain heart infusion (BHI) broth. Subsequently, cultured onto LB agar (Hi-Media, India), BHI agar ((Hi-Media, India), Nutrient agar (Hi-Media, India), Tryptic soy (TSA) agar (Hi-Media, India), MacConkey (Hi-Media, India) agar and Cetrimide

agar (Hi-Media, India) as selective media. Furthermore, suspected colonies were streaked on 10% Bovine Blood (BB) agar and incubation at 37˚C for 24 hours. Bacterial growth was initially confirmed by visually observing turbidity and then by measuring optical density (OD) using UV-Vis spectrophotometry. After 24h of incubation, the cultured broth was used to measure the OD, while the fresh LB, Nutrient broth, and BHI broth served as negative control (zero). The morphological and staining properties of the isolated bacteria were documented by the Gram's staining method through microscopic observations (100X), utilising a single pure colony obtained from a *P. aeruginosa* subculture. To confirm *P. aeruginosa*, indole synthesis, methyl red test, Voges-Proskauer test, hydrogen sulphide production, oxidase and catalase tests, and sugar fermentation tests were used. The Motility Indole Urease (MIU) test was also utilized to examine the organism's motility.

## Genomic DNA extraction and PCR amplification

Genomic DNA extraction from the isolates was performed through the boiling and thawing technique [38]. Subsequently, NanodropTM (Thermo Fisher Scientific, USA) was used for both quantifying and assessing the quality of the extracted DNA. To standardise the 16S rDNA oligonucleotides, bacterial DNA was mixed yielding an average concentration of 130 ng/μl and a purity of 2.10. The PCR reaction volume of 20 μl comprised 3 μl DNA template, 1 μl each of forward and reverse primers, 10 μl of PCR Master Mix, 2X (Promega, USA), and 5.5 μl of nuclease free water. Table 1 provides details on the primers and techniques used for the PCR reactions [39]. Analysis of the PCR end products was conducted using a 1.5% agarose gel. Post-electrophoresis, the gel was immersed in an ethidium bromide solution (concentration approximately 0.2–0.5 μg/mL) for 10 minutes and subsequently visualised using a UV transilluminator.

## Virulence profile of *P. aeruginosa*

The primer sets for PCR-based detection of the virulence-associated genes, along with the amplification protocols, were chosen from previously published papers with slight modifications, as summarized in **Table 1.** The visualization of PCR end products followed the procedure mentioned earlier.

## Antibiotic susceptibility of *P. aeruginosa*

The antibiogram test utilized commonly used antibiotics at the field level through the disc diffusion method [41]. In brief, the turbidity of all isolates was adjusted to a 0.5% McFarland standard by culturing them in LB broth (incubated at 37˚C for approximately 4 hours in a shaking incubator). Subsequently, the isolates were inoculated onto MHA (Liofilchem, Italy). Following inoculation, the antibiotic discs were placed and the entire setup was incubated at 37˚C for 18 hours to measure the diameter of the inhibition zone. To determine sensitivity or resistance of the isolates the Clinical and Laboratory Standards Institute (CLSI, 2019) standard interpretation chart for the zone of inhibition was referenced [42]. The antibiotic discs used in this study included Beta lactam Ampicillin G (AMP, 25 μg/disc), Ceftriaxone (CTR, 30 μg/disc), Cephradine (CE, 30 μg/disc), Meropenem (MEM, 10 μg/disc), Aztreonam (ATM, 30 μg/disc) Non-beta lactam Streptomycin (S, 10 μg/disc), Gentamycin (GEN, 30 μg/disc), Neomycin (N, 30 μg/disc), Doxycycline (DO, 30 μg/disc), Oxytetracycline (O, 30 μg/disc), Cotrimoxazole (COT, 25 μg/disc), Cefoxitin (FOX, 30 μg/disc), Chloramphenicol (C, 30 μg/disc), Florfenicol (FFC, 25 μg/disc), Nalidixic acid (NA, 30 μg/disc), Ciprofloxacin (CIP, 5 μg/disc), and Levofloxacin (LEV 5 μg/disc). Among the discs, CTR and GEN were purchased from HiMedia

**Table 1. List of primers, with oligonucleotide sequences, amplification conditions, amplicon size and references, which were used to detect *Pseudomonas* at genus and species level, as well as to detect the virulence genes by PCR.**

| Gene | Sequence (5'-3') | Thermal profile | | | | | | | | | | Amplicon size (bp) | References |
|------|------------------|-----------------|---|---|---|---|---|---|---|---|---|---------------------|------------|
| | | Initial denaturation | | Denaturation | | Annealing | | Elongation | | Final extension | | | |
| | | Tem | Time | Tem | Time | Tem | Time | Tem | Time | Tem | Time | | |
| 16S rDNA GS | F: GACGGGTGAGTAATGCCTA | 95°C | 2 m | 94°C | 20 s | 54°C | 20 s | 72°C | 40 s | 72°C | 10 m | 618 | [39] |
| | R: CACTGGTGTTCCTTCCTATA | | | | | | | | | | | | |
| 16S rDNA SS | F: GGGGGATCTTCGGACCTCA | 95°C | 2 m | 94°C | 20 s | 58°C | 20 s | 72°C | 40 s | 72°C | 10 m | 956 | |
| | R: TCCTTAGAGTGCCCACCCG | | | | | | | | | | | | |
| *exo*A | F: GACAACGCCCTCAGCATCACCAGC | 94°C | 2 m | 94°C | 1 m | 58°C | 1 m | 72°C | 30 s | 72°C | 5 m | 396 | [40] |
| | R: CGCTGGCCCATTCGCTCCAGCGCT | | | | | | | | | | | | |
| *exo*S | F: GCGAGGTCAGCAGAGTATCG | 94°C | 2 m | 94°C | 30 s | 58°C | 30 s | 72°C | 20 s | 72°C | 5 m | 118 | |
| | R: TTCGGCGTCACTGTGGAT | | | | | | | | | | | | |
| *exo*T | F: AATCGCCGTCCAACTGCATGCG | 94°C | 2 m | 94°C | 20 s | 58°C | 1 m | 72°C | 20 s | 72°C | 5 m | 152 | |
| | R: TGTTCGCCGAGGTACTGCTC | | | | | | | | | | | | |
| *exo*Y | F: CGGATTCTATGGCAGGGAGG | 94°C | 2 m | 94°C | 1 m | 58°C | 1 m | 72°C | 30 s | 72°C | 5 m | 289 | |
| | R: GCCCTTGATGCACTCGACCA | | | | | | | | | | | | |
| *exo*U | F: ATGCATATCCAATCGTTG | 94°C | 2 m | 94°C | 1 m | 54°C | 1 m | 72°C | 2 m | 72°C | 10 m | 2000 | |
| | R: TCATGTGAACTCCTTATT | | | | | | | | | | | | |
| *rhl*AB | F: TCATGGAATTGTCACAACCGC | 94°C | 2 m | 94°C | 20 s | 58°C | 20 s | 72°C | 15 s | 72°C | 7 m | 151 | |
| | R: ATACGGCAAAATCATGGCAAC | | | | | | | | | | | | |
| *rhl*R | F: CAATGAGGAATGACGGAGGC | 94°C | 2 m | 94°C | 1 m | 50°C | 30 s | 72°C | 45 s | 72°C | 7 m | 730 | |
| | R: GCTTCAGATGAGGCCCAGC | | | | | | | | | | | | |
| *rhl*I | F: CTTGGTCATGATCGAATTGCTC | 94°C | 2 m | 94°C | 1 m | 50°C | 30 s | 72°C | 40 s | 72°C | 7 m | 625 | |
| | R: ACGGCTGACGACCTCACAC | | | | | | | | | | | | |
| *las*I | F: ATGATCGTACAAATTGGTCGGC | 94°C | 2 m | 94°C | 1 m | 50°C | 30 s | 72°C | 35 s | 72°C | 7 m | 605 | |
| | R: GTCATGAAACCGCCAGTCG | | | | | | | | | | | | |
| *las*A | F: CGCCATCCAACCTGATGCAAT | 94°C | 2 m | 94°C | 1 m | 60°C | 1 m | 68°C | 30 s | 72°C | 7 m | 514 | |
| | R: AGGCCGGGGTTGTACAACGGA | | | | | | | | | | | | |

bp: Base pair; Tem = Temperature; m = minute; s = second; GS = genus-specific; SS = Species-specific.

Laboratories Private Limited (Maharashtra, India) and rest were from Oxoid Limited (Hampshire, UK).

## Statistical analysis

The data on location, age, gender, sample site, and test results, were first entered into Excel 2010 and then transferred to R 4.3.2 [43] for analysis. A quail bird was considered as affected if it tested positive for *Pseudomonas* spp. either in rectal or oral swabs. Continuous variable age was converted into categorical variable ($\leq$ 8 and > 8 weeks). The prevalence of *Pseudomonas* spp. across different variables was estimated using the 'tabpct' function of the R 'epiDisplay' package [44]. The association between *Pseudomonas* spp. infection and all explanatory variables was first investigated using univariable logistic regression model. Following that, multivariable logistic regression analysis was performed, incorporating only those explanatory variables with a univariable p-value of $\leq$ 0.2. Prior to conducting the multivariable logistic regression analysis, the variation inflation factor (VIF) was used to assess multicollinearity among the explanatory variables, with a cut-off level set at < 5. A backward model selection strategy was implemented for the final model selection, and the overall fit of the model was

evaluated using the Hosmer-Lemeshow test. The dataset used to identify risk factors for *P. aeruginosa* infection in quail birds is available in Supplementary File 1. The correlation coefficient between phenotypic resistance and the virulence profile was calculated using the 'cor' function in R version 4.3.2 [43].

## Sequencing and phylogenetic analysis

For the sequencing of the 16S rRNA gene, two isolates of *Pseudomonas* spp. (PCR detection up to genus level; 618 bp) were selected, ensuring coverage from both divisions (Dhaka and Mymensingh). Additionally, two isolates of *P. aeruginosa* (PCR detection at species level; 956 bp) were chosen from each division. The PCR was conducted using the aforementioned primer sets, reaction mixtures and thermal profiles. Successful amplifications were confirmed through 1.5% agarose gel electrophoresis and staining with ethidium bromide (EtBr) (Research-Lab Fine Chem Industries, Mumbai, 400 002, India), followed by visualization under a UV trans-illuminator (Biometra, Germany).

The amplified PCR products were purified using the QIAamp PCR product purification kit (QIAgen GmbH, Hilden, Germany). Subsequently, the purified PCR products underwent commercial sequencing at the National Institute of Biotechnology, Savar, Dhaka, Bangladesh, using the Sanger Dideoxy sequencing method. For sequencing a single amplicon, both forward and reverse primers were used. Raw sequence data were aligned, considering the forward sequence and the reverse complement of reverse sequence, and then edited using CLC sequence viewer v8 (http://www.clcbio.com) and BioEdit v7.2 [45] sequence editing and analysis tools. To determine sequence similarities with sequences already deposited in the NCBI GenBank, the edited sequences were subjected to BLAST. The GenBank accession numbers assigned to the present isolates are OR617385 to OR617388.

To assess the evolutionary relationships, a total of 56 partials and complete 16S rRNA gene sequences of *P. aeruginosa* were retrieved from NCBI GenBank. These sequences represent data from every division and source in Bangladesh (at least one from each division and source), neighbouring countries (India, Pakistan, China, Japan, Thailand) and at least one sequence from each continent. The 56 downloaded sequences, along with four of our sequences, were aligned using MUSCLE within MEGA11: Molecular Evolutionary Genetics Analysis version 11 software [46] package and exported in MEGA format. Finally, the MEGA11 software package was used to construct a phylogenetic tree employing 1000 Bootstrap replications. The tree was further annotated using the Interactive Tree Of Life (iTOL) v5: an online tool for phylogenetic tree display and annotation [47].

## Results

*Pseudomonas aeruginosa*, a significant bacterial pathogen, poses potential threats to both animal and human health. In this study, our primary objectives were to investigate the prevalence, antibiotic resistance patterns, and virulence gene profiles of *P. aeruginosa* in apparently healthy quail birds in Bangladesh. We aimed to explore potential risk factors associated with the presence of this pathogen.

## Phenotypic characteristics of the recovered isolates

The bacterial isolates showed robust growth in LB broth, characterized by dense turbidity and the formation of a pellicle. In nutrient broth at 37˚C for 24 hours under aerophilic conditions, the culture displayed turbidity with a green pigment, while BHI broth resulted in turbidity only. On LB agar, the organism demonstrated an earthy odour and irregular swarming growth. Conversely, on BHI agar, the isolates produced slimy mucoid colonies. Nutrient agar revealed

**Fig 1. Agarose gel electrophoresis results of PCR amplification targeting 16S rRNA gene for the genus- and species-specific detection of *Pseudomonas* spp. and *P. aeruginosa*, respectively.** (a) genus-specific PCR amplification of *Pseudomonas* spp. and positive bands appeared at 618 bp; (b) species-specific detection of *P. aeruginosa*, with amplicon size of 956 bp. In both cases, Lane M: 100 bp DNA ladder; NC: Negative control; Lane 1-9/10: Positive isolates.

small, irregular bluish-green colonies with a sweet odour. Cetrimide agar exhibited circular, smooth, small to medium-sized, mucoid greenish-blue opaque colonies. MacConkey agar displayed colourless, circular smooth, mucoid transparent colonies, while blood agar showed, irregularly edged circular flat smooth greyish non-haemolytic white colonies. Gram staining identified the organism as Gram-negative, rod-shaped, and arranged singly, in pairs, or in short chains. Furthermore, the Motility Indole Urease/Urea (MIU) medium test confirmed the organism's motility, but it tested negative for indole and urease. Suspected *Pseudomonas* spp. was found to be positive for Dextrose, Maltose, and Sucrose fermentation with acid and gas production, while testing negative for Lactose and Mannitol. Additionally, the organism exhibited a positive Methyl Red (MR) test but tested negative for the Indole and Voges-Proskauer tests.

## Confirmation by PCR

Genus level identification was confirmed by the presence of a positive band at 618 base pairs during gel electrophoresis (**Fig 1A**). Subsequently, specific detection of *P. aeruginosa* was confirmed by the presence of a 956 bp rRNA gene product (**Fig 1B**).

## Prevalence of *P. aeruginosa*

Of the 110 quail birds' rectal and oral swabs, cultural, Gram staining, and biochemical tests showed the presence of *Pseudomonas* spp. In 25 (22.73%) samples (Table 2). Specifically, *Pseudomonas* spp. were detected in 19.09% of rectal swabs, 6.36% of oral swabs, and 2.73% of samples obtained from both rectal and oral swabs (**Table 2**). Moreover, all isolates conventionally characterized were also confirmed positive in PCR amplification using specific primer sets. Fig 2 shows the district-level prevalence of *Pseudomonas* spp. infections.

## Risk factors analysis for isolated *P. aeruginosa*

The multivariable analysis incorporated four variables that showed associations with *P. aeruginosa* infection in quail. at p-values ≤0.2 in the univariable analysis (**Table 3**). The final multivariable logistic regression model identified three specific variables as risk factors (**Table 4**) for *P. aeruginosa* infection in quail. In Narsingdi, quail birds were found to be five times more likely to harbor *Pseudomonas* spp. (OR: 5.02; 95% CI:1.34–18.84) compared to those in Mymensingh Sadar. Quail birds younger than eight weeks had nearly six times higher odds (OR: 5.93; 95% CI: 1.96–17.91) of carrying *Pseudomonas* spp. compared to their older counterparts. Similarly, female quail birds showed almost four times higher odds of harbouring *Pseudomonas* spp. (OR: 3.77; 95% CI: 1.30–10.93) than male birds (**S1 Table**).

**Table 2. Prevalence of the isolated bacteria (culture and PCR-based).**

| District | Species | Sample type | Total samples | Total positive sample | Prevalence % | Overall prevalence |
|---|---|---|---|---|---|---|
| Mymensingh Sadar (n = 60) | Male | Oral swab | 30 | 0 (60) | 0 | Male = 9 (55), 16.36% Female = 16 (55), 29.09% |
| | | Rectal Swab | 30 | 2 (60) | 6.67 | |
| | Female | Oral swab | 30 | 3 (60) | 5 | |
| | | Rectal Swab | 30 | 6 (60) | 15 | |
| Jamalpur Sadar (n = 20) | Male | Oral swab | 10 | 0 (20) | 0 | |
| | | Rectal Swab | 10 | 2 (20) | 10 | |
| | Female | Oral swab | 10 | 1 (20) | 10 | |
| | | Rectal Swab | 10 | 2 (20) | 15 | |
| Narsingdi Sadar (n = 20) | Male | Oral swab | 10 | 1 (20) | 5 | |
| | | Rectal Swab | 10 | 3 (20) | 20 | |
| | Female | Oral swab | 10 | 1 (20) | 5 | |
| | | Rectal Swab | 10 | 2 (20) | 15 | |
| Gazipur Sadar (n = 10) | Male | Oral swab | 05 | 0 (10) | 0 | |
| | | Rectal Swab | 05 | 1 (10) | 10 | |
| | Female | Oral swab | 05 | 0 (10) | 0 | |
| | | Rectal Swab | 05 | 1 (10) | 10 | |
| 110 birds | | | **220** | **25** (110x2 = 220) | | **22.73% (25/110 birds)** |

## Distribution of virulence genes of *P. aeruginosa*

The presence of virulence genes among the 25 isolates (**S2 Table**) was as follows: *exo*A (100%; amplicon size 396 bp), *exo*S (88%; amplicon size 118 bp), *exo*T (64%; amplicon size 152 bp), *exo*Y (84%; amplicon size 289 bp), *rhl*AB (76%; amplicon size 151 bp), *rhl*R (100%; amplicon size 730 bp), *rhl*I (64%; amplicon size 625 bp). Notably, none of the isolates tested positive for *exo*U, *las*I and *las*A. The individual profile of virulence genes for each isolate is represented in **Table 5**, and the overall prevalence is visualized in **Fig 3**. Further, the PCR amplification results for each positive virulence gene is shown **Fig 4**.

## Antimicrobial susceptibility testing of *P. aeruginosa*

The analysis of the results of antibiotic resistance profile of *P. aeruginosa* was illustrated in the **Fig 4**. The results revealed that 92% of the isolates showed resistance to ampicillin, all isolates showed 100% resistance to 1[st] generation cephalosporin (Cephradine). Furthermore, resistance to other β-lactams varied, with cefoxitin displaying 76% resistance, while ceftriaxone and meropenem exhibited 100% susceptibility. Within the aminoglycoside group, streptomycin showed 28% susceptibility and 72% intermediate resistance. Conversely, Neomycin, Gentamycin, Cotrimoxazole, Chloramphenicol, Florfenicol, Ciprofloxacin, Levofloxacin demonstrated 100% susceptibility. On the other hand, the tetracycline group and quinolone group including Oxytetracycline, Doxycycline, and Nalidixic acid, showed 92% and 100% resistance respectively. The antibiotic resistance patterns for each isolate were showed in **Fig 5**. Moreover, **Fig 6** illustrated the isolate-wise distribution of virulence genes and the count of phenotypic antimicrobial resistance in *P. aeruginosa* isolates from quail birds in both the Mymensingh and Dhaka divisions.

## Phenotypic multi drug-resistance (MDR) nature of *P. aeruginosa* isolates

All *P. aeruginosa* isolates obtained from rectal and oral swabs of quail birds displayed a multi-drug-resistant (MDR) phenotype. The 25 isolates from both rectal and oral swabs exhibited

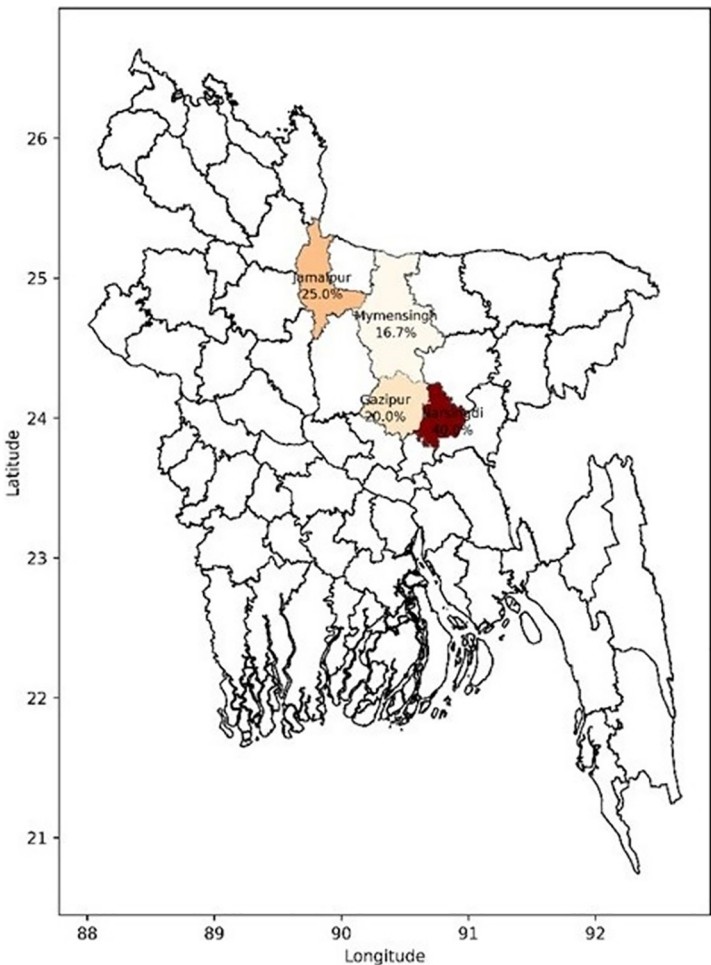

**Fig 2. District-level prevalence of *Pseudomonas* spp. infections across the study area.** The map was generated using Python with the Matplotlib and GeoPandas libraries. The district-level shapefile for Bangladesh was obtained from GADM (https://gadm.org/).

**Table 3. Univariable association between *Pseudomonas* spp. infection and explanatory variables.**

| Risk factors | Category level | *Pseudomonas* | | Prevalence (%) | OR | 95% CI | P-value |
|---|---|---|---|---|---|---|---|
| | | +ve | -ve | | | | |
| District | Mymensingh | 10 | 50 | 16.7 | Ref | - | - |
| | Jamalpur | 5 | 15 | 25.0 | 1.67 | 0.49–5.64 | 0.411 |
| | Gazipur | 2 | 8 | 20.0 | 1.25 | 0.23–6.79 | 0.800 |
| | Narsingdi | 8 | 12 | 40.0 | 3.33 | 1.08–10.25 | 0.036 |
| Sex | Male | 7 | 46 | 13.2 | Ref | - | - |
| | Female | 18 | 39 | 31.6 | 3.03 | 1.15–8.01 | 0.025 |
| Age (weeks) | > 8 | 9 | 57 | 13.6 | Ref | - | - |
| | ≤ 8 | 16 | 28 | 36.4 | 3.62 | 1.42–9.21 | 0.007 |
| Sample site | Market | 10 | 49 | 16.9 | Ref | - | - |
| | Farm | 15 | 36 | 29.4 | 2.04 | 0.82–5.06 | 0.124 |

+ve: Positive; -ve: Negative; OR: Odds ratio; CI: 95% confidence interval.

**Table 4. Risk factors identified in the final multivariable model for *Pseudomonas* spp. infection in quail birds.**

| Risk factors | Category level | OR | 95% CI | P-value |
|---|---|---|---|---|
| District | Mymensingh Sadar | Ref | - | - |
| | Jamalpur Sadar | 2.26 | 0.59–8.68 | 0.236 |
| | Gazipur Sadar | 3.01 | 0.47–619.50 | 0.247 |
| | Narsingdi Sadar | 5.02 | 1.34–18.84 | 0.017 |
| Sex | Male | Ref | - | - |
| | Female | 3.77 | 1.30–10.93 | 0.014 |
| Age (week) | > 8 | Ref | - | - |
| | < 8 | 5.93 | 1.96–17.91 | 0.002 |

OR: Odds ratio; CI: 95% confidence interval.

resistance to Aztreonam, Doxycycline, Chloramphenicol, Florfenicol, and Cefuroxime (ATM, DO, C, FFC, CXM), out of the 18 tested antibiotics. Among the 21 rectal swab isolates, nine exhibited resistances to a broader spectrum of nine antibiotics including Aztreonam, Doxycycline, Chloramphenicol, Florfenicol, Cefuroxime, Oxytetracycline, Ampicillin, Nalidixic acid, and Cefoxitin (ATM, DO, C, FFC, CXM, OT, AMP, NA, FOX,) (Table 5). In contrast, all 25 isolates were found to be susceptible to nine antibiotics, namely Meropenem, Ceftriaxone, Cephradine, Streptomycin, Neomycin, Gentamycin, Cotrimoxazole, Ciprofloxacin, and Levofloxacin. Though, all the 25 isolates were found as MDR, no isolate was detected as Extensively drug resistant (XDR) or pan drug resistant (PDR) (Table 5). A very weak positive linear relationship ($r = 0.11$) was observed between phenotypic resistance patterns and virulence profile (Fig 7).

The multiple antibiotic resistance index (MARI) analysis showed that all the isolates showed MARI values of more than 0.3 values, where PM11 and PD21 isolates had 0.352 (lowest among others), however, 13 isolates (PM1-2, PM5, PM7, PM9-10, PM16, PD17-19, and PD23-25) showed highest MARI (0.471). The rest 10 isolates had MARI value of 0.412 (Table 5).

## Sequencing and phylogenetic analysis

The phylogenetic analysis of the sequenced PCR amplification products revealed that, at the genus level, the *Pseudomonas* sp. isolates from quails in both Narsingdi (GenBank Accession No. OR617385) and Mymensingh (GenBank Accession No. OR617386) districts exhibited no evolutionary differences. Moreover, they displayed a close relationship with *Pseudomonas* isolates from milk (GenBank Accession No. MN464147; raw milk from Sylhet) and rhizosphere soil (GenBank Accession No. MG786551; source region not mentioned), respectively. Similarly, the analysed sequenced products of *P. aeruginosa* from quails in both Narsingdi (GenBank Accession No. OR617387) and Mymensingh (GenBank Accession No. OR617388) regions clustered very closely with the *P. aeruginosa* isolates from municipal waste in the Sylhet region (GenBank Accession No. MW040069) and hospital wastes from Khulna region (GenBank Accession No. MG735679), respectively (Fig 8).

## Discussion

Our findings highlight the prevalence of multi-drug-resistant and virulent potential *P. aeruginosa* in apparently healthy quail birds reared in Mymensingh and Dhaka divisions of Bangladesh. Young female birds should be regularly screened for future surveillance. We suggested a 'One Health' approach to proactively address and mitigate any future disasters caused by this organism in quail and human populations.

**Table 5. Frequency distribution of phenotypic resistance and virulence gene profiles of *P. aeruginosa* from rectal and oral swabs of quail birds with details antimicrobial categories and agents used to define MDR, XDR, PDR and MARI.**

| Isolate | Phenotypic resistance patterns against 17 antibiotics | | | | | | | | | | | | | | | | | | Virulence gens (% prevalence) | | | | | | |
|---|---|---|---|---|---|---|---|---|---|---|---|---|---|---|---|---|---|---|---|---|---|---|---|---|---|
| | C1 | C2 | | | C3 | C4 | C5 | | | C6 | | C7 | C8 | | C9 | | C10 | | *exo* | | | | *rhl* | | |
| | AMP | CTR | CE | FOX | MEM | ATM | S | GEN | N | DO | OT | COT | C | FFC | CIP | LEV | NA | MARI | A | S | T | Y | AB | I | R |
| R1 (PM1) | + | - | - | + | - | + | ± | - | - | + | + | - | + | + | - | - | + | 0.471 | √ | √ | √ | √ | √ | √ | √ |
| R6 (PM3) | + | - | - | - | - | + | ± | - | - | + | + | - | + | + | - | - | + | 0.412 | √ | √ | ● | ● | √ | √ | √ |
| R8 (PM4) | + | - | - | + | - | + | ± | - | - | + | + | - | + | + | - | - | - | 0.412 | √ | √ | √ | √ | √ | √ | √ |
| R7# (PM5) | + | - | - | + | - | + | ± | - | - | + | + | - | + | + | - | - | + | 0.471 | √ | √ | ● | √ | √ | √ | √ |
| R15# (PM6) | + | - | - | + | - | + | ± | - | - | + | + | - | + | + | - | - | - | 0.412 | √ | √ | √ | √ | ● | ● | √ |
| R22 (PM8) | + | - | - | - | - | + | ± | - | - | + | + | - | + | + | - | - | + | 0.412 | √ | √ | ● | √ | √ | ● | √ |
| R44 (PM9) | + | - | - | - | - | + | ± | - | - | + | + | - | + | + | - | - | + | 0.471 | √ | √ | ● | ● | √ | ● | √ |
| R56 (PM10) | + | - | - | + | - | + | ± | - | - | + | + | - | + | + | - | - | + | 0.471 | √ | ● | √ | √ | √ | √ | √ |
| R57 (PM11) | - | - | - | + | - | + | - | - | - | + | + | - | + | + | - | - | - | 0.352 | √ | √ | √ | √ | ● | ● | √ |
| R61 (PM12) | + | - | - | + | - | + | ± | - | - | + | - | - | + | + | - | - | + | 0.412 | √ | √ | ● | √ | √ | ● | √ |
| R63 (PM13) | + | - | - | - | - | + | - | - | - | + | + | - | + | + | - | - | + | 0.412 | √ | √ | √ | √ | √ | √ | √ |
| R65 (PM14) | + | - | - | + | - | + | ± | - | - | + | + | - | + | + | - | - | - | 0.412 | √ | √ | √ | √ | √ | ● | √ |
| R71 (PM15) | + | - | - | + | - | + | - | - | - | + | + | - | + | + | - | - | + | 0.412 | √ | √ | √ | √ | √ | √ | √ |
| R82 (PD18) | + | - | - | + | - | + | ± | - | - | + | + | - | + | + | - | - | + | 0.471 | √ | √ | √ | √ | √ | √ | √ |
| R84 (PD19) | + | - | - | + | - | + | ± | - | - | + | + | - | + | + | - | - | + | 0.471 | √ | ● | √ | ● | √ | √ | √ |
| R86# (PD20) | + | - | - | - | - | + | ± | - | - | + | + | - | + | + | - | - | + | 0.412 | √ | √ | √ | √ | ● | ● | √ |
| R88 (PD21) | + | - | - | + | - | + | ± | - | - | + | - | - | + | + | - | - | - | 0.352 | √ | √ | √ | √ | √ | ● | √ |
| R91 (PD22) | + | - | - | - | - | + | ± | - | - | + | + | - | + | + | - | - | + | 0.412 | √ | √ | ● | √ | √ | √ | √ |
| R100 (PD23) | + | - | - | + | - | + | ± | - | - | + | + | - | + | + | - | - | + | 0.471 | √ | ● | ● | ● | ● | ● | √ |
| R103 (PD24) | + | - | - | + | - | + | ± | - | - | + | + | - | + | + | - | - | + | 0.471 | √ | √ | √ | √ | √ | √ | √ |
| R107 (PD25) | + | - | - | + | - | + | ± | - | - | + | + | - | + | + | - | - | + | 0.471 | √ | √ | ● | √ | ● | √ | √ |
| O3 (PM2) | + | - | - | + | - | + | ± | - | - | + | + | - | + | + | - | - | + | 0.471 | √ | √ | √ | √ | ● | √ | √ |
| O7# (PM5) | + | - | - | + | - | + | ± | - | - | + | + | - | + | + | - | - | + | 0.471 | √ | √ | ● | √ | ● | ● | √ |
| O15# (PM6) | + | - | - | + | - | + | ± | - | - | + | + | - | + | + | - | - | - | 0.412 | √ | √ | √ | √ | ● | ● | √ |
| O19 (PM7) | - | - | - | + | - | + | - | - | - | + | + | - | + | + | - | - | + | 0.471 | √ | √ | √ | √ | √ | √ | √ |
| O72 (PM16) | + | - | - | + | - | + | - | - | - | + | + | - | + | + | - | - | + | 0.471 | √ | √ | √ | √ | √ | √ | √ |
| O81 (PD17) | + | - | - | + | - | + | ± | - | - | + | + | - | + | + | - | - | + | 0.471 | √ | √ | ● | √ | √ | √ | √ |
| O86# (PD20) | + | - | - | - | - | + | ± | - | - | + | + | - | + | + | - | - | + | 0.412 | √ | √ | √ | √ | ● | ● | √ |
| % resistant | 92 | - | - | 76 | - | 100 | - | - | - | 100 | 92 | - | 100 | 100 | - | - | 80 | | 100 | 88 | 64 | 84 | 76 | 64 | 100 |

**Note:** *O = Oral swabs, R = Rectal swabs, # Samples positive for both samples, + = Resistant,— = Sensitive, ± = Intermediately sensitive, C = Class of antibiotic, PM = *Pseudomonas* isolate from Mymensingh division, PD = *Pseudomonas* isolate from Mymensingh division, PM1-11: Mymensingh district, PM12-16: Jamalpur district, PD17-23: Narsingdi district, PD24-25: Gazipur district, % = percentage.

**C1**: Beta lactam [Ampicillin AMP], **C2**: Cephalosporin [Ceftriaxone (CTR), Cefuroxime (CXM), Cephradine (CE), Cefoxitin (FOX)], **C3**: Carbapenem [Meropenem (MEM)], **C4**: Monobactums [Aztreonam (AT)], **C5**: Aminoglycosides [Streptomycin (S), Gentamycin (GEN), Neomycin (N)], **C6**: Tetracycline [Doxycycline (DO), Oxytetracycline (O)], **C7**: Sulfonamides [Cotrimoxazole (COT)], **C8**: Phenicols.

[Chloramphenicol (C), Florfenicol (FFC)], **C9**: Fluoroquinolones [Ciprofloxacin (CIP), Levofloxacin (LEV)], **C10**: Quinolones [Nalidixic acid (NA)]

√ = indicate presence, ● = indicate absence, *exo*U, *las*I and *las*A were not found, so excluded from this table

Criteria for defining MDR, XDR and PDR in *Pseudomonas aeruginosa*: MDR: non-susceptible to ≥1 agent in ≥3 antimicrobial categories, XDR: non-susceptible to ≥1 agent in all but ≤2 categories, PDR: non-susceptible to all antimicrobial agents listed.

http://www.ecdc.europa.eu/en/activities/diseaseprogrammes/ARHAI/Pages/public_consultation_clinical_microbiology_infection_article.aspx

# All the 25 isolates were found as MDR, but no XDR and PDR isolate was observed.

*P. aeruginosa* infection in quail birds poses a potential public health threat due to the zoonotic nature of this pathogen. The antibiotic resistance, pathogenicity, and transmission behaviour of *P. aeruginosa* are growing concern in livestock and public health, and

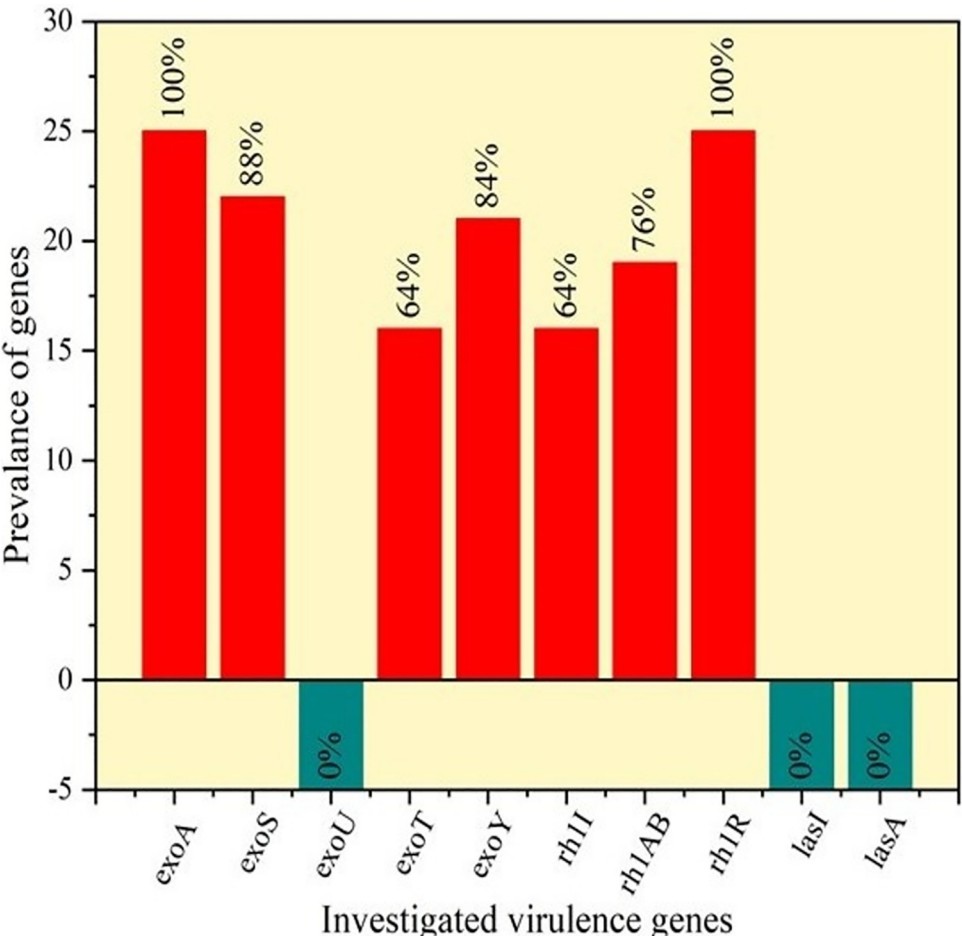

**Fig 3. Frequency distribution of virulence genes in isolated *P. aeruginosa*.**

environmental health [48]. Therefore, studying *Pseudomonas* infections in quail birds offers valuable insights into pathogenicity, genetic variation, and resistance patterns, which can help inform effective treatment and prevention strategies. To address this issue, the present study was designed to provide a detailed analysis of the virulence and antibiogram profiling of *P. aeruginosa* isolated from apparently healthy Japanese quail (*Coturnix japonica*) in Bangladesh.

The susceptibility of young birds to *P. aeruginosa* infections is well-documented, with the potential for mechanical transmission through skin injuries or contaminated needles used in vaccinations. Factors such as the bird's immunological status and concurrent illnesses can increase their vulnerability [49]. This study identified PCR positive isolates of *P. aeruginosa*, in quails at a proportion of 22.73%. Notably, there is a scarcity of studies on *P. aeruginosa* in poultry in Bangladesh. A recent study by Bakheet and Torra (2020) reported a 24% positivity in poultry [50], while another study found a 28.3% prevalence in broiler chickens from Ismailia Governorate, Egypt [51]. A higher prevalence (34%) of *P. aeruginosa* was reported in broiler chickens [52], while a 20% prevalence was observed in broiler chickens using 16S rDNA [53]. In this study, the prevalence of *P. aeruginosa* in quail birds from Narsingdi district was significantly higher than in Mymensingh. This variation may be due to differences in management practices, agro-climatic conditions such as rainfall, humidity, and temperature, as well as the local prevalence of *P. aeruginosa* in the Narsingdi region. The impact of regional factors as a

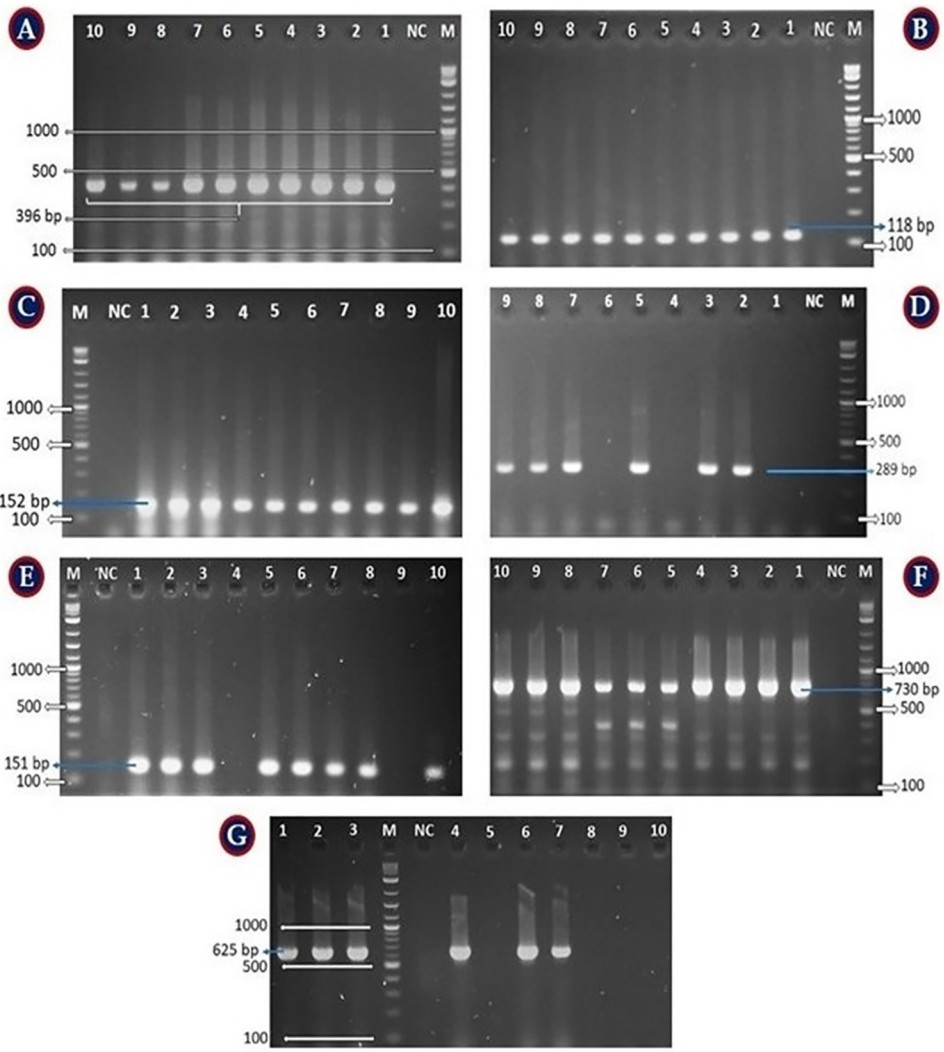

**Fig 4. Virulence genes detection in PCR confirmed *P. aeruginosa* isolates.** [A] *exo*A (396 bp), [B] *exo*S (118 bp), [C] *exo*T (152 bp), [D] *exo*Y (289 bp), [E] *rhl*AB (151 bp), [F] *rhl*R (730 bp), and [G] *rhl*I (625 bp). However, there were no *exo*U, *las*I and *las*A found in any isolates. In all cases, the Lane M = 100 bp DNA ladder; NC = negative control; and clear and consistent band showing lanes indicated positive isolates.

risk factor aligns with findings from human infections, where regional variations also contribute to differences in prevalence [54,55].

In terms of age, the study found that the prevalence of *P. aeruginosa* in young birds (<8 weeks old) was six times higher than in older birds (>8 weeks old). This may occur because young birds have an underdeveloped immune system, which limits their ability to produce specific antibodies and T cells, as well as a lower number of phagocytic cells. As a result, their immune system struggles to effectively recognize and eliminate pathogens like *P. aeruginosa* [56,57]. Additionally, nutritional deficiencies and immature mucosal surfaces increase the permeability and susceptibility of young birds, facilitating the colonization of *P. aeruginosa*. This observation is consistent with existing literature [31,57,58]. These studies highlighted the heightened susceptibility of young chicks to *P. aeruginosa* infections. Additionally, mortality was noted to be higher in young birds [59], further emphasizing the vulnerability of this age group.

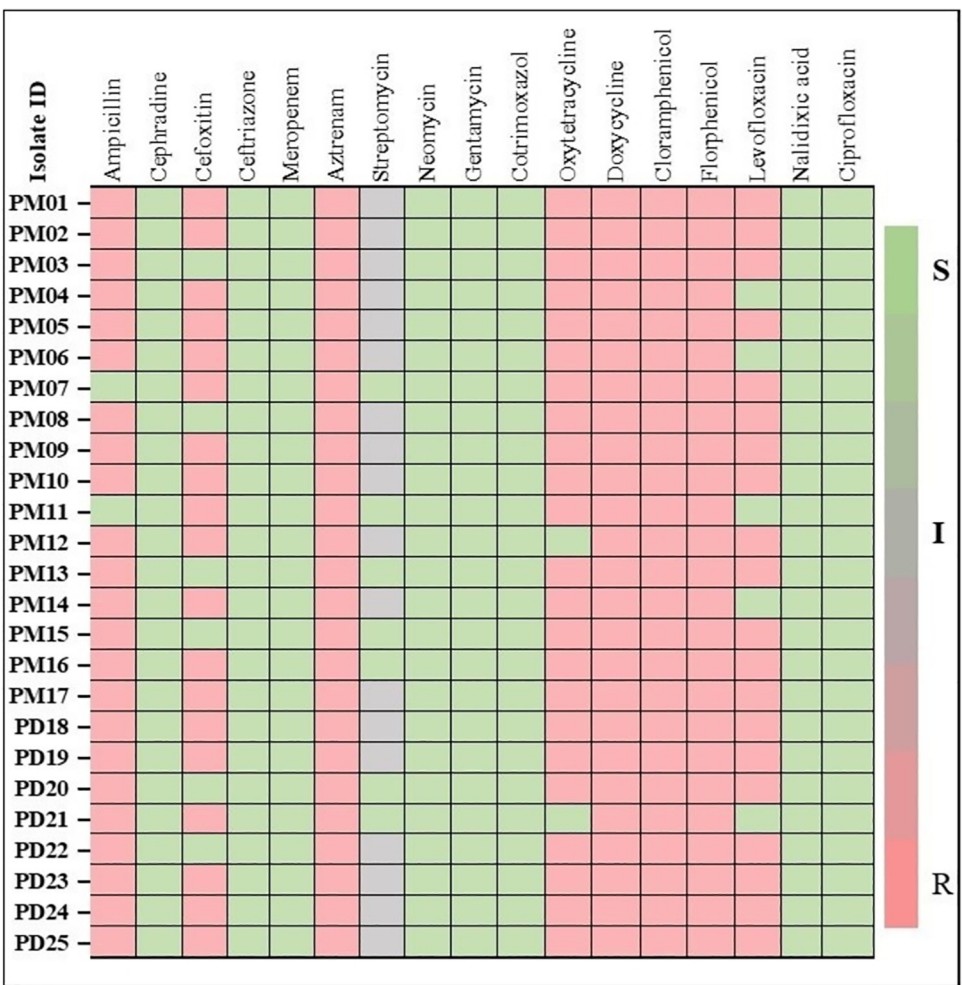

**Fig 5. Isolate-wise antibiotic resistance patterns of the isolated *P. aeruginosa* from oral and rectal swabs of quail birds.**

Furthermore, the study identified a higher prevalence of *P. aeruginosa* in female birds compared to male birds This finding aligns with studies on human infections, where female patients are more susceptible than males [60,61] The female reproductive hormones may contribute to increased vulnerability to *P. aeruginosa* infection [62].

Understanding the pathophysiology of opportunistic pathogens such as *P. aeruginosa* is important and characterizing the virulence factors that constitute its defensive mechanisms is instrumental in exploring novel antimicrobial strategies, especially for multidrug-resistant (MDR) strains. *P. aeruginosa* secretes various extracellular components to support its survival and enhance virulence [63,64]. The four effector proteins, ExoS, ExoT, ExoU, and ExoY, secreted by *P. aeruginosa* through the type III secretion system, play a pivotal role in its pathogenicity [65]. These proteins impact critical host cell activities, including signal transduction and cytoskeletal organization [66]. Wareham and Curti [67] described ExoS and ExoT as bifunctional toxins capable of activating GTPase and ADP-ribosyltransferase activities. In this study, we identified the virulence genes *exo*S (88%) and *exo*T (64%), consistent with their findings, which noted that ExoT exhibits lower ADP-ribosyltransferase activity than ExoS [67]. Additionally, ExoU, known for its phospholipase activity, causes eukaryotic membrane

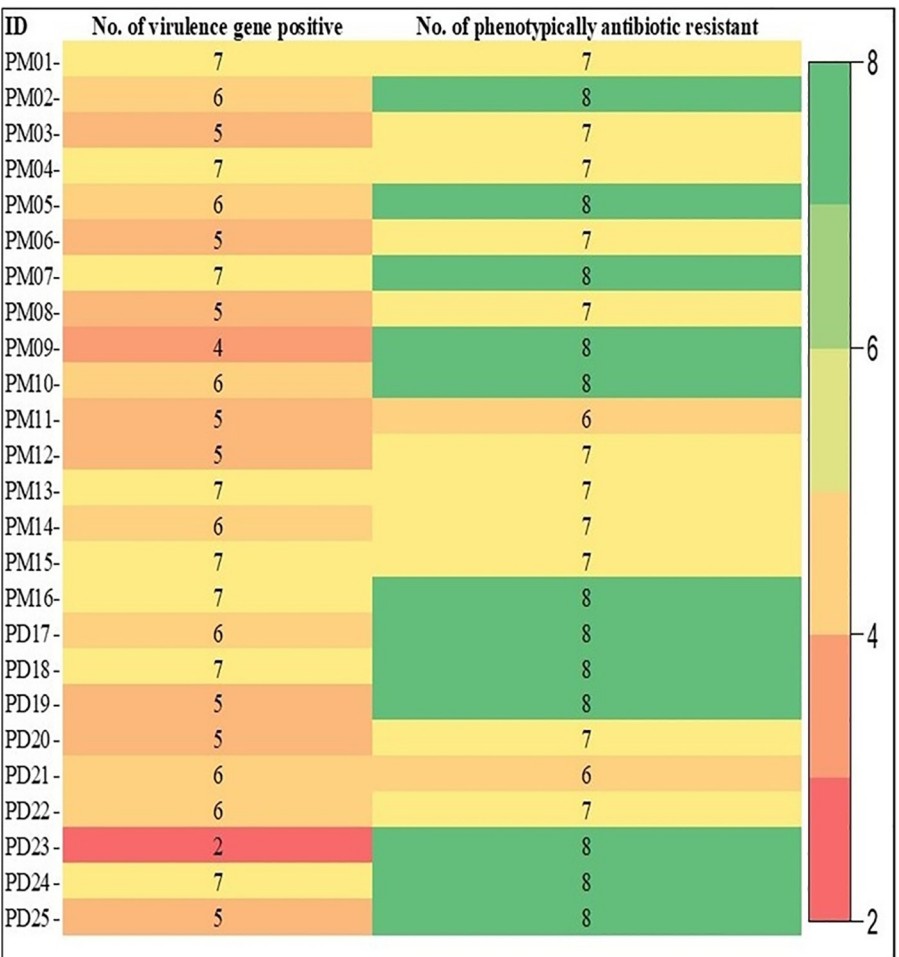

**Fig 6. Isolate-wise with total number-based virulence genes occurrence and phenotypic antimicrobial resistance in the *P. aeruginosa* isolates.**

rupture upon entering the cytoplasm, while ExoY possesses adenylate cyclase activity. The presence of *exo*Y (84%) in our study suggests that *P. aeruginosa* exhibited adenylate cyclase activity, whereas *exo*U was not detected. Furthermore, the cytotoxin protein synthesis gene, *exo*A, was found in 100% of the isolates, a result similar to Ghazaei [68], who reported *exo*A in 83.33% of animal samples. The quorum sensing (QS) systems of *P. aeruginosa*, particularly the conventional LuxI/R-type systems—LasI/R and RhlI/R—play a crucial role in regulating virulence factor synthesis and biofilm formation. According to Bakheet and Torra [50], these systems generate and detect 3OC12-homoserine lactone and C4-homoserine lactone, respectively. Interestingly, the *las*I gene, responsible for secreting key virulence factors such as pyocyanin and elastase, was not detected in any isolates in this study, which contrasts with Bratu et al. [69], who reported the presence of *las*I in 100% of *P. aeruginosa* strains. Similarly, *las*R was present in 75% of strains in studies by Kebede [70] and Saleh et al. [71]. Despite the absence of the *las*I gene, QS activity was confirmed in this study through the detection of *rhl*R (100%), *rhl*I (64%), and other markers. Furthermore, *P. aeruginosa* can develop antibiotic resistance through mutations in its DNA or via horizontal gene transfer. The presence of plasmids, integrons, and transposons carrying antibiotic-resistant genes in *P. aeruginosa* is well documented [55]. The multidrug-resistant characteristics of *P. aeruginosa*, coupled with its

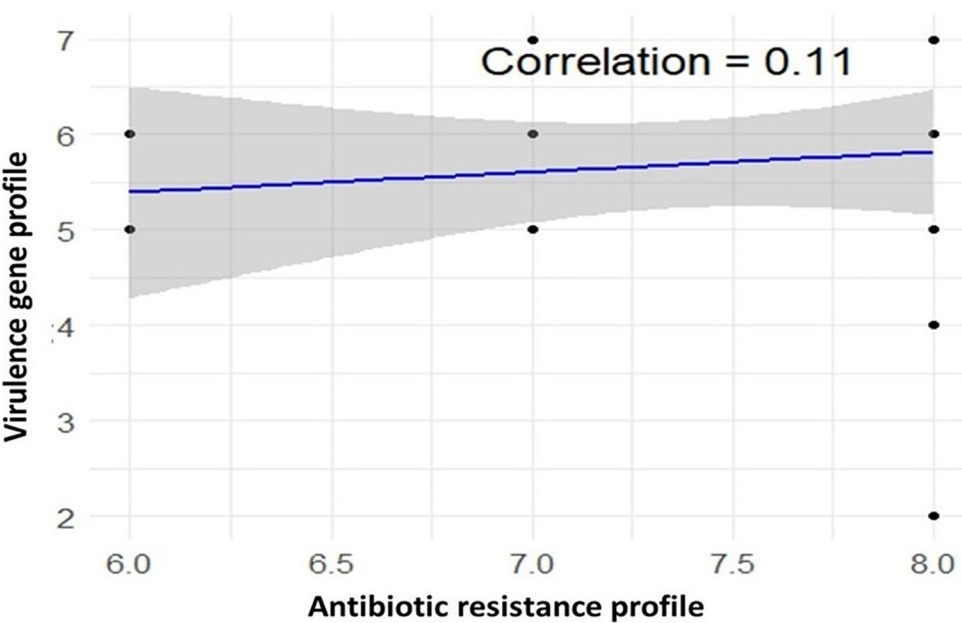

**Fig 7. Correlation between phenotypic resistance patterns and virulence profiles.**

potential for transmission to humans and animals, make it a significant public health concern [57].

Notably, 92% of isolates displayed resistance to ampicillin, while all isolates exhibited 100% resistance to first-generation cephalosporin (Cephradine). These findings align with those of Shahat et al. [53] and are consistent with reports from other studies, which also identified high levels of resistance to Tetracycline and Ampicillin [72–74].

Interestingly, within the β-lactam category, cefoxitin exhibited 76% resistance, while ceftriaxone and meropenem showed complete susceptibility (100%). These results contrast with the general resistance trend observed in the β-lactam group, highlighting the nuanced response of *P. aeruginosa* to different antibiotics within the same class. In the aminoglycoside group, streptomycin displayed 28% susceptibility and 72% intermediate resistance. However, neomycin, gentamycin, cotrimoxazole, chloramphenicol, florfenicol, ciprofloxacin, and levofloxacin all demonstrated 100% susceptibility. These findings align with Shahat et al. [54] and support the effectiveness of ciprofloxacin and gentamycin against *P. aeruginosa*, as noted by Mohammad [75]. Conversely, the tetracycline and quinolone groups, including oxytetracycline, doxycycline, and nalidixic acid, showed 92% and 100% resistance, respectively. The high resistance to nalidixic acid (80%) is consistent with observations by [76]. Additionally, it was noted that 96% of isolates had Multiple Antibiotic Resistance Index (MARI) values greater than 0.2, indicating a high-risk source of antibiotic-resistant bacterial contamination [77]. These findings closely align with the report by Afunwa et al. [78].

## Conclusion

This study highlights the concerning prevalence of *Pseudomonas aeruginosa* in quail birds. Younger and female quail from Narsingdi district should be prioritized for future surveillance and control measures. Infected birds should be treated with antibiotics to which *P. aeruginosa* is susceptible. The consistent presence of virulence genes *exo*A and *rhl*R, combined with the genetic diversity observed in other virulence markers, underscores the pathogenic potential of

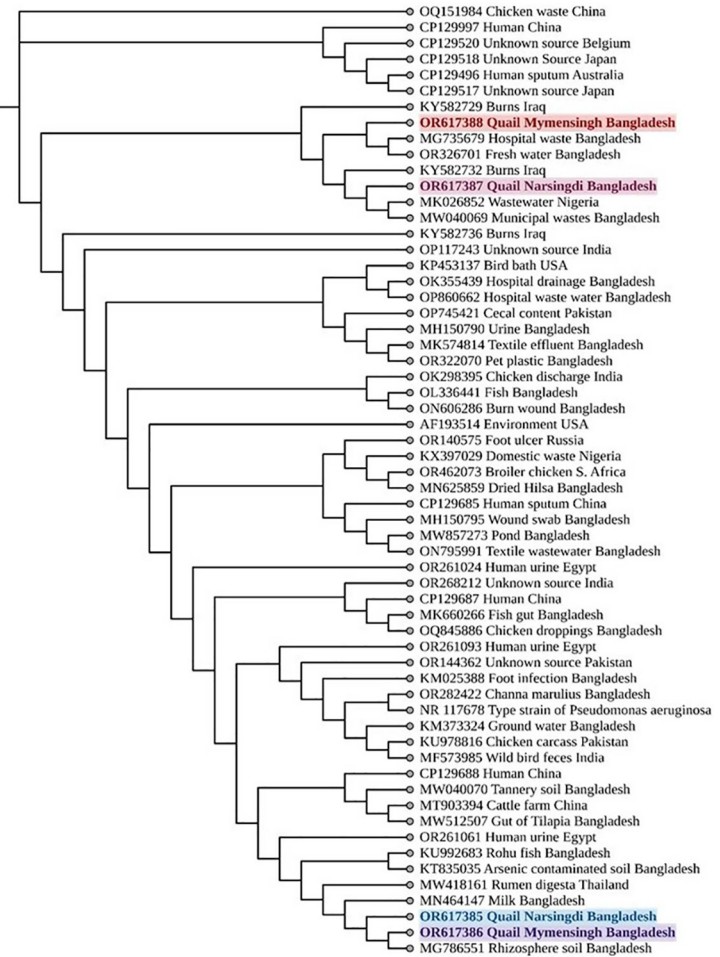

**Fig 8. Phylogenetic relationship of the isolated *P. aeruginosa* from quail birds of Narsingdi and Mymensingh regions of Bangladesh.** The sequenced and analysed isolates were marked by various colour.

the isolates. These findings emphasize the need for further research into the epidemiology, virulence profiling, and genetic diversity of *P. aeruginosa* in quail, which is crucial for developing effective preventive strategies, including vaccines.

## Supporting information

**S1 Table. Gender wise positive distribution of *P. aeruginosa* at different areas.**
(DOCX)

**S2 Table. Sample wise positive distribution of different virulence genes of *P. aeruginosa*.**
(DOCX)

## Author Contributions

**Conceptualization:** Mahbubul Pratik Siddique.

**Data curation:** Alamgir Hasan, Md. Tanjir Ahmmed, Bushra Benta Rahman Prapti.

**Formal analysis:** A. K. M. Anisur Rahman, Mahbubul Pratik Siddique.

**Investigation:** Alamgir Hasan, Md. Tanjir Ahmmed, Mahbubul Pratik Siddique.

**Methodology:** Alamgir Hasan, Md. Tanjir Ahmmed, Bushra Benta Rahman Prapti, Mahbubul Pratik Siddique.

**Supervision:** Mahbubul Pratik Siddique.

**Validation:** A. K. M. Anisur Rahman, Mahbubul Pratik Siddique.

**Visualization:** Aminur Rahman, Tasnim Islam, Chandra Shaker Chouhan.

**Writing – original draft:** Alamgir Hasan, Md. Tanjir Ahmmed.

**Writing – review & editing:** Alamgir Hasan, Md. Tanjir Ahmmed, Bushra Benta Rahman Prapti, Aminur Rahman, Tasnim Islam, Chandra Shaker Chouhan, A. K. M. Anisur Rahman, Mahbubul Pratik Siddique.

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
