## [Decision Letter · Decision Letter 0]

18 Oct 2024

PONE-D-24-41634First Time Detail Virulence and antibiogram profiling of Pseudomonas aeruginosa isolated from apparently healthy Japanese quail (Coturnix japonica) in BangladeshPLOS ONE

Dear Dr. Siddique,

Thank you for submitting your manuscript to PLOS ONE. After careful consideration, we feel that it has merit but does not fully meet PLOS ONE’s publication criteria as it currently stands. Therefore, we invite you to submit a revised version of the manuscript that addresses the points raised during the review process.

We look forward to receiving your revised manuscript.

Kind regards,

Morteza Saki

Academic Editor

PLOS ONE

Journal Requirements:

“CONFLICT OF INTEREST

19th September 2024

Title: First Time Detail Virulence and antibiogram profiling of Pseudomonas aeruginosa isolated from apparently healthy Japanese quail (Coturnix japonica) in Bangladesh

DECLARATION AND STATEMENT

We know of no conflicts of interest associated with the publication. We declare that this manuscript is original and has not been submitted to anywhere else for publication. The manuscript nicely falls under the scope of your journal.

Sincerely yours,

Mahbubul Pratik Siddique, PhD

Corresponding author

and

Professor

Department of Microbiology and Hygiene

Bangladesh Agricultural University

Mymensingh-2202

Cell Phone: +880-1751-945123

Email: mpsiddique@bau.edu.bd

     https://www.researchgate.net/profile/Mahbubul-Siddique

     https://scholar.google.com/citations?user=YmxVEzEAAAAJ&hl=en”

3. We note that your Data Availability Statement is currently as follows: [All relevant data are within the manuscript and its Supporting Information files]

6. We note that [Figure 1] in your submission contain [map/satellite] images which may be copyrighted. All PLOS content is published under the Creative Commons Attribution License (CC BY 4.0), which means that the manuscript, images, and Supporting Information files will be freely available online, and any third party is permitted to access, download, copy, distribute, and use these materials in any way, even commercially, with proper attribution. For these reasons, we cannot publish previously copyrighted maps or satellite images created using proprietary data, such as Google software (Google Maps, Street View, and Earth). For more information, see our copyright guidelines: http://journals.plos.org/plosone/s/licenses-and-copyright.

Reviewers' comments:

Reviewer's Responses to Questions

**Comments to the Author**

1. Is the manuscript technically sound, and do the data support the conclusions?

Reviewer #1: Yes

Reviewer #2: Yes

2. Has the statistical analysis been performed appropriately and rigorously? 

Reviewer #1: Yes

Reviewer #2: Yes

3. Have the authors made all data underlying the findings in their manuscript fully available?

Reviewer #1: Yes

Reviewer #2: Yes

4. Is the manuscript presented in an intelligible fashion and written in standard English?

Reviewer #1: No

Reviewer #2: Yes

5. Review Comments to the Author

Reviewer #1: Comments to authors:

-The current study is interesting; however, the authors should address the following comments to improve the quality of the manuscript:

-The manuscript should be revised for English editing and grammar mistakes.

-Please write the scientific names of bacterial pathogens and genes in the correct form all over the manuscript and the references section.

Title:

I think the work would benefit from the title that contains the main conclusion of the study (should be derived from the conclusion). Please modify the title.

Abstract:

- The abstract must illustrate the used methods and the most prevalent results (give more hints about methods and results). Besides, rephrase the aim of the work and the main conclusion of your findings.

- Add the full expression before the abbreviations.

-Introduction: (it needs to be more informative):

-Give a hint about the virulence factors and the mechanism of disease occurrence , and infecions caused by P. aeruginosa.

-Illustrate the mechanism of action different virulence factors of P. aeruginosa.

-Rephrase the aim of the work to be clear and better sound.

Material and methods:

-Support all methods with updated specific references.

•Add the company, city, and country of the used chemicals and reagents.

-Isolation and identification:

Discuss in detail the methods of isolation and identification of P. aeruginosa. Besides, specific references should be added.

•Add the company, city, and country of the used bacterial media and reagents that were used in the biochemical identification of isolates. Also, enumerate all used biochemical reactions.

-PCR based detection of other antimicrobial resistance genes should be performed. Afterwards, the correlation between phenotypic and genotypic multidrug resistance should be performed.

-MLST should be carried out to illustrate the genetic relatedness among the retrieved isolates (If available).

-Results:

-Please add a starting paragraph to the results section to briefly introduce the topic, your goals and hypothesis and a short summary of what you did in this work.

-Antimicrobial susceptibility testing:

• Illustrate in a new table the occurrence of MDR (Multidrug resistance) among the recovered isolates as the following (illustrate the names of the antimicrobial classes and different antibiotics):

No. of strains%Type of resistance

R, MDR, and XDRPhenotypic multidrug resistance

(Antimicrobial classes and different antibiotics).The antibiotic-resistance genes

-Tables 6 and 7: Address the multidrug resistance papperns in the tables. I couldn't find it (MDR, XDR, and PDR)

-MLST should be carried out to illustrate the genetic relatedness among the retrieved isolates.

-Support your findings with more illustrating figures. Besides, increase the resolution of all figures (must be 600 dpi).

-Discussion:

-The authors are advised to illustrate the real impact of their findings without repetition of results.

-Please illustrate the role of virulence genes and virulence determinants of P. aeruginosa in the pathogenesis of the disease.

-Please illustrate different mechanisms of antibacterial resistance P. aeruginosa.

-Illustrate the essential role of biofilm in the occurence of infection.

-Conclusion

-Should be rephrased to be sounded. A real conclusion should focus on the question or claim you articulated in your study, which resolution has been the main objective of your paper?

Reviewer #2: Dear Editor:-

I think this article is very good regarding the originality of the idea at which the field of study is so interesting and medically important.

The topic of manuscript is timely and will be of interest to the readers of the journal. Also, the manuscript is very well written and the ideas flow logically.

Dear authors you are doing well and that’s look great work. A few issues, however, need to be addressed;

In line 13 page 2 Abstract: Abstract was too long, must be 250 words.

In line 13 page 2 Give a brief explanation of the significance of Pseudomonas aeruginosa for bird health and its effects on the poultry trade. This might facilitate the setting of your research.

In line 17 page 2 Provide details on how the quail were selected for sampling. Also, mention whether the birds were healthy or showing signs of illness.

What effects do the virulence genes (exoA, rhlR, etc.) that have been found have on P. aeruginosa's pathogenicity in quail? Could you explain more about the relevance of other genes' varied distribution?

Is there a biological or environmental reason why female quail are more likely than male quail to host Pseudomonas spp.?

In line 28 page2 What biological or environmental factors could explain the increased odds of female quail harboring Pseudomonas spp. compared to males?

In line 35 page2 Given that exoU, lasI, and lasA were not detected in any isolates, how might this influence the virulence potential of these strains?

In line 84 page 5 Regarding P. aeruginosa from quails, how does the rise in multidrug resistance (MDR) correspond to developments in resistance in human medicine?

In line 88 page 5 To enhance introduction section add the following reference: Rawaa A. Hussein, Shaymaa H. AL-Kubaisy, Mushtak T.S. Al-Ouqaili. The influence of efflux pump, outer membrane permeability and β-lactamase production on the resistance profile of multi, extensively and pandrug resistant Klebsiella pneumoniae. Journal of Infection and Public Health, Volume 17, Issue 11, 2024. https://doi.org/10.1016/j.jiph.2024.102544.

In line 90 page 5 Why are quail farms mentioned in this particular setting? Could you explain on how common AMR is, particularly in quail farms as opposed to other livestock?

In line 100 page 6 To enhance introduction section add the following reference: Alduhaidhawi AHM, AlHuchaimi SN, Al-Mayah TA, Al-Ouqaili MTS, Alkafaas SS, Muthupandian S, Saki M. Prevalence of CRISPR-Cas Systems and Their Possible Association with Antibiotic Resistance in Enterococcus faecalis and Enterococcus faecium Collected from Hospital Wastewater. Infect Drug Resist. 2022 Mar 19;15:1143-1154. doi: 10.2147/IDR.S358248. PMID: 35340673; PMCID: PMC8942119.

In line 118 page 7 What guidelines were applied when choosing the particular areas for the sample—Jamalpur, Mymensingh, Gazipur, and Narsingdi? Do these regions reflect the larger quail farming sector in Bangladesh?

In line 119 page 7 Why were oral and cloacal swabs the only ones used to get samples? Would other sample types—like environmental or fecal samples—offer more thorough insights?

In line 122 page 7 Why are samples from male and female quails being collected in equal numbers, and why specifically? Regarding illness susceptibility or AMR prevalence, were gender variations taken into account?

In line 149 page 8 On one hand authors worked with boiling and thawing for DNA extraction technique, but why Ethidium bromide was used. Ethidium bromide- a carcinogen!

In line 153 page 8 What is the clinical significance of detecting the specified virulence-associated genes in your study? How do the presence of these genes correlate with pathogenicity or virulence in Pseudomonas?

In line 232 page 14 In what ways was turbidity assessed or quantified in your cultures? When evaluating the turbidity levels in LB broth, nutritional broth, and BHI broth, did you employ any particular methods or tools?

In line 267 page 16 Results by sample type and gender are shown in Table 2. Why are the prevalence statistics for male and female birds presented differently, in your opinion? What knowledge does this analysis of gender differences offer?

In line 273 page 16 What underlying variables could be responsible for the higher probability of Pseudomonas spp. being present in quail from Narsingdi as opposed to Mymensingh Sadar?

In line 290 page 17 The absence of genes such as exoU, lasI, and lasA is noteworthy. What implications does this have for the virulence potential of these isolates? Could their absence indicate a particular environmental adaptation?

In line 293 page 17 Has any genetic diversity study been done on the 25 isolates of P. aeruginosa? In what ways could genetic variation affect the way virulence genes are distributed?

In line 308 page 19 It is remarkable that 72% of specimens have intermediate streptomycin resistance. In the setting of treatment, how should this be understood, and what kind of follow-up testing is recommended?

In line 326page 20 Nine of the 18 antibiotics evaluated in the study demonstrated susceptibility. How did these antibiotics come to be chosen? Were they chosen because they were commonly used antibiotics in clinical settings, or because they were relevant to veterinary medicine?

In line 326page 23 Could you provide more details about how your research on quail birds in particular advances our knowledge of P. aeruginosa's potential harm to public health? In what way does this study contribute to the overall understanding of P. aeruginosa epidemiology?

In line 326page 24 To enhance discussion section add the following reference: Murshid RM, Al-Ouqaili MTS, Kanaan BAJ. Microbial Vaginosis and its Relation to Single or Multi-Species Biofilm in Iraqi Women: Clinical and Microbiological Study. Pak J Biol Sci. 2024 Jul;27(8):404-412. doi: 10.3923/pjbs.2024.404.412. PMID: 39300677.

In line 388 page 24 According to result of this research, the likelihood of contracting P. aeruginosa is six times higher in young birds (less than 8 weeks old). In younger birds, what particular biochemical or immunological pathways could explain this heightened vulnerability? Has the literature put forth or examined any plausible theories?

In line 401page 24 To enhance discussion section add the following reference: •Saleh RO, Al-Ouqaili MTS, Ali E, Alhajlah S, Kareem AH, Shakir MN, Alasheqi MQ, Mustafa YF, Alawadi A, Alsaalamy A. lncRNA-microRNA axis in cancer drug resistance: particular focus on signaling pathways. Med Oncol. 2024 Jan 9;41(2):52. http://doi.org/ 10.1007/s12032-023-02263-8.

In line 450 page 27 Conclusion should be objective with further perspective or should add at least a few sentences about future study/future perspective of it

6. PLOS authors have the option to publish the peer review history of their article (what does this mean?). If published, this will include your full peer review and any attached files.

Reviewer #1: No

Reviewer #2: No

---

## [Author Response · Author response to Decision Letter 0]

27 Nov 2024

Edditor 

Comments to authors and RESPONSE

Manuscript ID: PONE-D-24-41634

Title: First Time Detail Virulence and antibiogram profiling of Pseudomonas aeruginosa isolated from apparently healthy Japanese quail (Coturnix japonica) in Bangladesh

Modified title: First Time MDR Virulent Potential Pseudomonas aeruginosa from Apparently Healthy Japanese quail (Coturnix japonica) in Bangladesh

Response: We are thankful to the editor and reviewers for their valuable and constructive comments, which helped us improve the quality of our manuscript. We have carefully revised the manuscript considering their suggestions and believe that the revised version meets the expectations of the editor and reviewers. Our responses to the reviewers’ specific comments are provided in red coloured text in the revised manuscript.

Journal Requirements:

3. We note that your Data Availability Statement is currently as follows: [All relevant data are within the manuscript and its Supporting Information files]

Response: The data generated during this study is included in the manuscript and its associated supplementary files. 

4. PLOS ONE now requires that authors provide the original uncropped and unadjusted images underlying all blot or gel results reported in a submission’s figures or Supporting Information files. 

Response: We thank to the editor for this comment. The raw blot/gel images are included in the Figure section not to Supporting Information. 

Response: We thank to the editor for this comment. Sorry for such inconvenience and we have removed “data not shown” from the manuscript accordingly. 

6. We note that [Figure 1] in your submission contain [map/satellite] images which may be copyrighted. All PLOS content is published under the Creative Commons Attribution License (CC BY 4.0), which means that the manuscript, images, and Supporting Information files will be freely available online, and any third party is permitted to access, download, copy, distribute, and use these materials in any way, even commercially, with proper attribution. For these reasons, we cannot publish previously copyrighted maps or satellite images created using proprietary data, such as Google software (Google Maps, Street View, and Earth). For more information, see our copyright guidelines: http://journals.plos.org/plosone/s/licenses-and-copyright.

Response: Thanks to the editor for this comment. Now, we have produced Figure 2 in Python using Matplotlib and Geopandas libraries. We have downloaded the district level Bangladesh shape file from GADM (https://gadm.org/) 

Reviewer #1: 

Comments to authors and RESPONSE

-The current study is interesting; however, the authors should address the following comments to improve the quality of the manuscript:

-The manuscript should be revised for English editing and grammar mistakes.

Response: Thanks to the reviewer for this comment. We have critically checked the manuscript for English editing and fixing grammar mistakes.

-Please write the scientific names of bacterial pathogens and genes in the correct form all over the manuscript and the references section.

Response: We thank the reviewer for this comment. We have italicized the scientific names of bacterial pathogens and genes throughout the manuscript and in the reference section.

Title:

I think the work would benefit from the title that contains the main conclusion of the study (should be derived from the conclusion). Please modify the title.

Response: We have modified the title as “First Time MDR Virulent Potential Pseudomonas aeruginosa from apparently healthy Japanese quail (Coturnix japonica) in Bangladesh”.

Abstract:

- The abstract must illustrate the used methods and the most prevalent results (give more hints about methods and results). Besides, rephrase the aim of the work and the main conclusion of your findings.

Response: We have modified the abstract according to the suggestions of the reviewer. 

- Add the full expression before the abbreviations.

Response: Thank you. We have elaborated before using abbreviated form.

-Introduction: (it needs to be more informative):

-Give a hint about the virulence factors and the mechanism of disease occurrence, and infections caused by P. aeruginosa.

Response: According to the reviewer’s comments, we have added/updated the virulence factors, the mechanism of disease occurrence and infections caused by P. aeruginosa. 

-Illustrate the mechanism of action different virulence factors of P. aeruginosa.

Response: The mechanism of action of different virulence factors had already been described in the submitted manuscript.

-Rephrase the aim of the work to be clear and better sound.

Response: The aims of the works have been rephrased to improve readability and clarity.

Material and methods:

-Support all methods with updated specific references.

Response: All methods have been cited with appropriate references.

•Add the company, city, and country of the used chemicals and reagents.

Response: The company, city and country of the used chemicals and reagents have been added.

-Isolation and identification:

Discuss in detail the methods of isolation and identification of P. aeruginosa. Besides, specific references should be added.

Response: Modified according to the suggestions of the reviewer.

•Add the company, city, and country of the used bacterial media and reagents that were used in the biochemical identification of isolates. Also, enumerate all used biochemical reactions.

Response: The company, city and country of the used bacterial media and reagents have been added.

-PCR based detection of other antimicrobial resistance genes should be performed. Afterwards, the correlation between phenotypic and genotypic multidrug resistance should be performed.

Response: In our study, we did not detect antimicrobial resistance genes among the Pseudomonas aeruginosa isolates. However, we did identify virulence genes and conducted a correlation analysis between the phenotypic resistance profiles and the virulence characteristics of the isolates. Please see Table 5 and Figure 7. 

-MLST should be carried out to illustrate the genetic relatedness among the retrieved isolates (If available).

Response: We appreciate the insightful suggestion from the reviewer regarding the use of MLST to assess genetic relatedness between the recovered isolates. While we acknowledge the value of MLST in providing detailed insights into bacterial population structures, unfortunately, due to resource constraints, we were unable to conduct MLST analysis in this study. Our primary focus was on investigating the prevalence, risk factors, virulence, and antimicrobial resistance profiles of P. aeruginosa in apparently healthy quail birds. We believe that the comprehensive data presented in our study, encompassing both phenotypic and genotypic aspects, contributes significantly to understanding the potential public health implications associated with P. aeruginosa in quail populations.

-Results:

-Please add a starting paragraph to the results section to briefly introduce the topic, your goals and hypothesis and a short summary of what you did in this work.

Response: We have added a starting paragraph to briefly introducing the topic, our goal, hypothesis and a short summary of what have done.

-Antimicrobial susceptibility testing:

• Illustrate in a new table the occurrence of MDR (Multidrug resistance) among the recovered isolates as the following (illustrate the names of the antimicrobial classes and different antibiotics):

No. of strains%Type of resistance R, MDR, and XDRPhenotypic multidrug resistance

(Antimicrobial classes and different antibiotics). The antibiotic-resistance genes

-Tables 6 and 7: Address the multidrug resistance patterns in the tables. I couldn't find it (MDR, XDR, and PDR)

Response: Cordial thank and gratitude again to the honourable reviewer for this kind comments and useful suggestion. We have added a new table (Table 5 in the revised version) illustrating the details phenotypic resistance and virulence profiling, indicating their MARI, MDR, XDR, PDR, their percent prevalence. Due to repetition of same theme, we have deleted Table 5 of previous version. 

MLST should be carried out to illustrate the genetic relatedness among the retrieved isolates.

Response: Thank you again. We already added the explanation in the Materials and Methods section of the Reviewer’s comments. We should explore and incorporate this striking feature in our future researches, if resources are available.

-Support your findings with more illustrating figures. Besides, increase the resolution of all figures (must be 600 dpi).

Response: We have improved the resolution of all figures (at least 600 dpi)

Moreover, for more clarification, even we changed figure 2, 3, 5, and 6 with new look and higher resolution

-Discussion:

-The authors are advised to illustrate the real impact of their findings without repetition of results.

Response: We have discussed the real impact of our findings without repeating the results.

-Please illustrate the role of virulence genes and virulence determinants of P. aeruginosa in the pathogenesis of the disease.

Response: We have discussed different mechanisms of antibacterial resistance in P. aeruginosa.

-Please illustrate different mechanisms of antibacterial resistance P. aeruginosa.

Response: We have discussed different mechanisms of antibacterial resistance in P. aeruginosa.

-Illustrate the essential role of biofilm in the occurrence of infection.

Response: We have discussed different mechanisms of antibacterial resistance in P. aeruginosa. 

-Conclusion

-Should be rephrased to be sounded. A real conclusion should focus on the question or claim you articulated in your study, which resolution has been the main objective of your paper?

Response: We have rewritten the conclusion as suggested.

Reviewer #2: 

Comments to authors and RESPONSE

: Dear Editor:-

I think this article is very good regarding the originality of the idea at which the field of study is so interesting and medically important.

The topic of manuscript is timely and will be of interest to the readers of the journal. Also, the manuscript is very well written and the ideas flow logically.

Dear authors you are doing well and that’s look great work. A few issues, however, need to be addressed;

In line 13 page 2 Abstract: Abstract was too long, must be 250 words.

Response: We thank the reviewer for this comment. The abstract has been revised and shortened to 300 words in accordance with the journal guidelines.

In line 13 page 2 Give a brief explanation of the significance of Pseudomonas aeruginosa for bird health and its effects on the poultry trade. This might facilitate the setting of your research.

Response: We thank the reviewer for this comment. The significance of Pseudomonas aeruginosa for bird health and its impact on the poultry industry has been incorporated into the abstract, as mentioned in the previous comment.

In line 17 page 2 Provide details on how the quail were selected for sampling. Also, mention whether the birds were healthy or showing signs of illness.

Response: The abstract has been revised in accordance with the reviewer’s suggestions, as mentioned in the previous comment.

What effects do the virulence genes (exoA, rhlR, etc.) that have been found have on P. aeruginosa's pathogenicity in quail? Could you explain more about the relevance of other genes' varied distribution.?

Response: We thank the reviewer for this comment. The presence of virulence genes such as exoA, rhlR, exoS, exoY, and rhlI encodes toxins, enzymes, and various secretory proteins that facilitate bacterial persistence, colonization, survival, and infection of the host under adverse conditions. These genes contribute to biofilm formation, induce quorum sensing, promote cell death, cause tissue damage, and manipulate host cellular processes. The specific roles of these virulence genes have been incorporated into the introduction section as suggested.

Is there a biological or environmental reason why female quail are more likely than male quail to host Pseudomonas spp.?

Response: We thank the reviewer for this comment. Hypothetically, we propose that female quail may be more susceptible to certain Pseudomonas species due to hormonal influences, differences in immune responses, and behavioral patterns. Additionally, genetic predispositions, including differences in gene expression related to immune function, could play a role in increasing susceptibility in females compared to males. These factors may explain why female quail are more likely to host *Pseudomonas* spp. than their male counterparts.

In line 28 page2 What biological or environmental factors could explain the increased odds of female quail harboring Pseudomonas spp. compared to males?

Response: We thank the reviewer for this comment. Several factors may contribute to the increased odds of female quail harboring Pseudomonas spp. compared to males. From a biological perspective, female quail secrete sex hormones such as estrogen and progesterone, which can influence immune function and potentially enhance susceptibility to infections by modulating immune responses. Additionally, differences in the gut and skin microbiomes between females and males may create conditions more favorable for bacterial colonization in females. From an environmental perspective, female quail might be more frequently exposed to environments with a higher prevalence of Pseudomonas spp., leading to increased contact with contaminated substrates. These combined factors could explain the observed difference in susceptibility.

In line 35 page 2 Given that exoU, lasI, and lasA were not detected in any isolates, how might this influence the virulence potential of these strains?

Response: We thank the reviewer for this comme

---

## [Decision Letter · Decision Letter 1]

16 Dec 2024

First Report of MDR Virulent Pseudomonas aeruginosa in Apparently Healthy Japanese Quail (Coturnix japonica) in Bangladesh

PONE-D-24-41634R1

Dear Dr. Mahbubul Pratik Siddique,

We’re pleased to inform you that your manuscript has been judged scientifically suitable for publication and will be formally accepted for publication once it meets all outstanding technical requirements.

Kind regards,

Morteza Saki

Academic Editor

PLOS ONE

Additional Editor Comments (optional):

Reviewers' comments:

Reviewer's Responses to Questions

**Comments to the Author**

1. If the authors have adequately addressed your comments raised in a previous round of review and you feel that this manuscript is now acceptable for publication, you may indicate that here to bypass the “Comments to the Author” section, enter your conflict of interest statement in the “Confidential to Editor” section, and submit your "Accept" recommendation.

Reviewer #1: All comments have been addressed

Reviewer #2: All comments have been addressed

2. Is the manuscript technically sound, and do the data support the conclusions?

Reviewer #1: Yes

Reviewer #2: Yes

3. Has the statistical analysis been performed appropriately and rigorously? 

Reviewer #1: Yes

Reviewer #2: Yes

4. Have the authors made all data underlying the findings in their manuscript fully available?

Reviewer #1: Yes

Reviewer #2: Yes

5. Is the manuscript presented in an intelligible fashion and written in standard English?

Reviewer #1: Yes

Reviewer #2: Yes

6. Review Comments to the Author

Reviewer #1: The authors have carried out significant changes to the manuscript. They have addressed most of the suggested corrections and comments. Really, it's an interesting study that has a significant impact. Now, the manuscript could be accepted.

Congratulations.

Reviewer #2: Dear author

You are doing well. The manuscript now is so arranged and more acceptable for publication

Regard

7. PLOS authors have the option to publish the peer review history of their article (what does this mean?). If published, this will include your full peer review and any attached files.

Reviewer #1: No

Reviewer #2: **Yes: **Mushtak T.S. Al-Ouqaili

---

## [Editor Report · Acceptance letter]

15 Jan 2025

PONE-D-24-41634R1 

PLOS ONE

Dear Dr. Siddique, 

I'm pleased to inform you that your manuscript has been deemed suitable for publication in PLOS ONE. Congratulations! Your manuscript is now being handed over to our production team.

Kind regards, 

on behalf of

Dr. Morteza Saki 

Academic Editor

PLOS ONE